# The Use of Live Action, Animation, and Computer-Generated Imagery in the Depiction of Non-Human Primates in Film

**DOI:** 10.3390/ani12121576

**Published:** 2022-06-18

**Authors:** Alexandra Martinez, Marco Campera, K. A. I. Nekaris

**Affiliations:** 1Nocturnal Primate Research Group, School of Social Sciences, Oxford Brookes University, Oxford OX3 0BP, UK; alexandramj7@gmail.com (A.M.); anekaris@brookes.ac.uk (K.A.I.N.); 2Department of Biological and Medical Sciences, Oxford Brookes University, Oxford OX3 0BP, UK

**Keywords:** CGI, animal actors, movies, animal welfare, conservation, public perception

## Abstract

**Simple Summary:**

Films offer a rare opportunity to share messages with a wide audience, although the messages provided can have negative effects on animal conservation, such as direct negative effects on animal welfare, increased illegal trade, and negative percepts among the public. Here we aim to understand how the medium used in film can impact the films gross profit worldwide and film critic consensus scores using 101 English-speaking films that portrayed any primate that debuted between 1 January 2000 and 31 December 2019. We found that films with computer-generated imagery (CGI) primates were more likely to have higher profits and critic scores. We suggest that the use of the CGI medium can positively impact the general public perception of primates, and this has the potential to create positive implications for conservation.

**Abstract:**

For over 100 years, non-human primates (primates) have been a part of the now hundred-billion-dollar global film industry in a variety of capacities. Their use in the film industry is of concern due to the negative welfare effects on individuals, the potential for increased pet trade, and the conservation impacts of public perception. While the effects on human perception of using live primates in film have been studied, little research has been performed on their appearance in animation and none in computer-generated imagery (CGI). We aimed to investigate how the portrayal of primates varied between depiction medium types and how this related to the films’ performance with critics and in the box office. We observed 151 primates in 101 different English-speaking films that debuted between 2000 and 2019. For each appearance we recorded aspects of primate portrayals based on accuracy, anthropomorphism, environment, and agency displayed, along with the depiction medium. We used structural equation models to depict the highest likelihood of the portrayal aspects on the medium’s relationship to the films gross profit worldwide and film critic consensus scores. We found that over the 20-year time frame, use of live primates has decreased, CGI has increased, and animations have remained relatively steady. While animation had no significant relationship to gross profit or critic consensus, both were significantly lower for films that used live primates and were significantly higher for films that used CGI primates. Due to the steady increase in the use of the CGI medium and its positive relationship with gross profit and critic consensus, it could have great effects on people’s perceptions of primates and implications for conservation efforts.

## 1. Introduction

In 2019, the global motion picture industry market surpassed 100 billion United States dollars (USD) [1]. While film is a major economic industry, it is also a reflection of human culture and a lens through which we see ourselves [2,3,4]. With that, animals and animal imagery have been used by the film industry in all genres, time periods, and with every major technological development [5,6]. Non-human primates (primates hereafter), in particular, have long captured the interest of storytellers and filmmakers. For example, the 1918 film *Tarzan of the Apes* has had nearly 50 sequels and remakes in 100 years. Early 20th century producers Cooper and Schoedsack also exemplified this by heavily featuring primates alongside humans in films, culminating with their blockbuster *King Kong* (1933). Primates have since played many roles on screen, such as monsters, pets, protagonists, and props, portrayed using different media including live animals, animation, computer-generated imagery (CGI), and even humans in costume.

The use of live-action animals in film has been prevalent since the inception of motion pictures [5], and though animal performers were nothing new, their use on screen became an extremely lucrative business [6]. Animals remained expendable for entertainment throughout the 1930s, with a pivotal moment in *Jesse James* (1939) where a real horse was run off a cliff to its death, after which the public sought reforms [5,6]. These days, despite increasing efforts for legislation to phase out the use of exotic animals in film and media due to welfare concerns, it remains a large part of the film industry [7].

No medium has become synonymous with leading animal characters quite like animation. The author Wells [8] reasons that this is due to the nature of its arduous production along with the social role that animation came to hold. Initially, instead of the intricacies of drawing humans, the exaggerated features and simple lines of anthropomorphized animals made it easier to produce the abundance of frames needed to create a film. This led to the ability to create fantastical and comical relief from people’s everyday lives in the approaching financial crisis [8]. In many ways, especially evident in animation, animals on film are at once merely animals and not quite animals [5,8]. They often face very human issues, with human voices, and human expression, but with the ever-present allusion to their animal ‘reality’. This duality of animated animals can strike connections with viewers, adding empathy for and interest in the presented species [9,10,11]. However, there is some concern for the emphasis on their aesthetics over accuracy that has been a topic most often discussed in the illustrations of animals in children’s books [12,13], but warrants further discourse in film [14]. Though animation may also appear to be a better medium to depict exotic animals, over having live-action ‘actor’ animals, the conclusion that animal welfare is never directly impacted in the making of the film would be wrong.

In the last 40 years, an observed shift in the types of films gracing the top grossing spots in the US showed a tendency to be more technologically advanced films [15]. Using CGI, creators are able to import imagery into their computer or develop it there and then freely manipulate it to their will and needs [16]. This more recent film technology is most impactful because it allows content creators to produce stunning images and feats with less time and expense than live action or animation [15]. The greatest advancement of CGI in films has been this freedom to manipulate what appears as real in unrealistic and fantastical ways [16]. The criticism that CGI faces most is that seems too real. CGI animals have been shown to elicit the same uneasy feeling and cognitive dissonance as hyper-realistic computer-generated humans, known as the ‘uncanny valley’ [17].

The cultural and economic power of film, paired with its long-intertwined history of animal exploitation, creates obvious concerns for those in biodiversity conservation. Yet, what attracts filmmakers and viewers to primates in particular are often the same reasons that their conservation is important. As our closest relatives, we continue to study wild primates for a greater understanding of our own species [18]. They also play an important role in cultural traditions within their natural range countries [18]. Furthermore, they function as key components of those regions’ ecology [19], but due to anthropogenic forces, around 75% of primates have declining populations [18]. However, due to their long life histories and specialized nature, primates also take longer to recover from population decline than other animals [20]. Currently, the effect of films on conservation has been explored in three major areas: its direct effect on animal welfare, influence on wildlife trade, and relationship to public knowledge and perceptions [2,21,22,23,24,25].

Primates, like other exotic animals, have intricate specialized diets, complex social systems, and can pose a serious danger to all people they come in contact with, whether they are trained or not [26]. The American Humane Association provides notes to film producers on these issues, but that has not deterred all use of live primates, nor does it recognize all negative welfare implications [27]. For example, many films show primates using a bared-teeth display, which by humans is often interpreted as a smile, but in many primates is a signal of fear, discomfort, or submission [28]. Not only is teeth display a concern for the individual psychological welfare as it may indicate that the animal is in distress, but these misconceptions may impact the view that primates and other animals make suitable and friendly pets, increasing the likelihood of trade [29,30,31].

Of all live mammals that are trafficked globally, primates and carnivores make up the largest percentage [32]. In a previous review of primate ‘actors’ in film, chimpanzees (*Pan troglodytes*) and capuchins (*Cebus* spp. And *Sapajus* spp.) made the most appearances [33]. While there are multiple capuchin species that are considered Endangered or Critically Endangered by the International Union for Conservation of Nature and Natural Resource (IUCN), none are protected under the US Endangered Species Act (ESA). On the other hand, chimpanzees are a special case under the ESA; until the 2015 final ruling, captive chimpanzees were not considered Endangered like their wild conspecifics, thus their trade was not protected [34].

Laws regarding trade in primates and their welfare needs can be convoluted and confusing for the general public, while their portrayals in film may be where people gain the most exposure to many exotic species. Thus, appearance and perception greatly affect the future welfare and trade of primates. For chimpanzees in particular, the context in which people view them affects their perceived threat status [21,23,35]. When viewed with a human, subjects were less likely to identify the species as Endangered and more likely to identify them as suitable pets [21]. Just like other globally threatened taxa, perceived need is the most influential motive for conservation donors [36,37]. Thus, for globally threatened species to be assumed safe based on media portrayal can hinder conservation efforts [29,30]. Anthropomorphism has seen to be an effective tool to introduce unfamiliar species to the public and aid in empathy for the species; however, it is suggested that animals are given ‘just enough’ anthropomorphic characteristics to make them credible social actors [11]. Despite this, there is a risk of anthropomorphism overshadowing accuracy [11]. Animals’ roles in film can affect their conservation whether they appear in live action, CGI or animation, through welfare, trade, and general public information. The images in terms of accuracy and anthropomorphism can also have a substantial effect on people’s perceptions, thus in turn affecting species conservation.

While the effects of live primates in the film industry have been a major focus for conservationists, few have studied the effect when portrayed using various media, such as animation or CGI. To understand how the film industry impacts primate conservation, we must first understand the types of portrayals being cultivated into the cultural consciousness. The anthropomorphism, accuracy, and environmental context in which primates appear have all shown to influence how people understand the primates and the threats that face them. Yet, each major medium—live action, animation, and CGI—in which they appear holds its own set of challenges. Thus, how primates are depicted by each medium may differ, as well as each medium’s relationship with audience reception. Understanding this relationship may assist filmmakers when forming decisions that could affect the animals they depict and their films’ success. With this study we seek to determine the following: if the depiction medium (medium hereafter) affects audience reception through critic evaluation and gross film profit; how the portrayal in each medium differs in accuracy, anthropomorphism, environment, and agency; and to characterize the interplay between audience reception and primate portrayal.

## 2. Materials and Methods

### 2.1. Film List Compilation

We compiled a list of English-speaking feature length films that portrayed any primate that debuted between 1 January 2000 and 31 December 2019, due to the relatively recent development of CGI. With the continuing rise of online streaming-exclusive films, the list was not restricted by traditional theatre release, and included films released exclusively online by popular steaming services, such as Netflix.

To compile the films, we first consulted the Primate Info Net’s list of primates in film and recorded all films mentioned within the time frame to an initial list [38]. In addition, we consulted the International Movie Database (IMDb) list of films released in each year. We accessed the first four pages of the list for every year, beginning with the top grossing films, each page consisting of 50 films. We first put films that featured a primate on the promotional poster on the initial list, and then set aside any film set in a primate range country for further review. We similarly consulted Wikipedia’s annual American movie release list to cross reference the Primate Info Net and IMDb lists and ensure a wide range of included films. We then searched the set aside films to identify the possibility of primate presence in two ways. First, we watched trailers via YouTube and IMDb, and moved films with primates onto the initial list. Second, we researched the film synopses and used a text search to find the presence of keywords, ‘monkey’, ‘lemur’, ‘baboon’, ‘ape’, ‘chimp’, or ‘gorilla’ in the summary. We included all films debuting between 2000 and 2019, and that contained a primate in the promotional poster, trailer, or synopsis in the final list.

### 2.2. Data Collection

We classified each film by the genus or genera depicted, as many films displayed multiple genera of primates in different capacities. In the case of an unrecognizable genus, we consulted the film’s online cast or production materials to determine if there was an intended species representation. If the cursory search did not reveal a specific genus, we categorized the primate as an unidentifiable generic monkey or generic ape. For each film we recorded the title, release year, director, Classification and Rating Administration (CARA) rating, and listed genres. The CARA rating includes G (General Audiences), PG (Parental Guidance Suggested), PG-13 (Parents Strongly Cautioned), and R (Restricted). We recorded audience-reception data as continuous variables, and all others as dichotomous, with 1 being present and 0 being absent. A.M. performed the scoring. The only groups that were mutually exclusive were prop compared to agency, no vocalization compared to anthro-like vocalization or natural-like vocalization, and the variables of the living environment group (Table A1).

The primate medium was not always representative of the film’s main medium. For example, *Mary Poppins Returns* in 2019 was a live-action film with animated inserts, and all primates depicted in that film were animated, and thus included in the animated category. The medium was also not mutually exclusive, due to some films using mixed mediums to depict the same animal. However, CGI live action was separated from CGI animation by degree of attempted realism. For example, *Wonder Park* (2019) is a CGI animated film with cartoon-like characters and categorized as animation, whereas The Lion King (2019) is a CGI live-action film and thus categorized as simply CGI.

Between 12 June 2020 and 29 July 2020, we watched all the films on the final list to assess the primates shown. We accessed all the films via online the streaming services Disney+, Netflix, Hulu, Amazon Prime Video, and Xfinity on Demand, or recorded them from Xfinity television broadcasts. While watching the films, we recorded all data in Excel 2016 and added every primate genus depicted in each film upon appearance, even if some occurrences were not initially expected (Appendix A).

### 2.3. Data Analysis

We used IBM SPSS Statistics 26 to run frequencies and descriptive statistics. We calculated the means and range for USD worldwide, IMDb audience scores, Rotten Tomatoes audience scores, and Rotten Tomatoes critic scores. We calculated the frequencies of the CARA ratings, human interaction, and primate genera frequency differentiated by medium.

We conducted an exploratory series of path-analysis SEMs using IBM SPSS AMOS 26 graphics [39]. The Structural Equation Model (SEM) is a theory-based analysis technique often used in psychology and other social sciences to test a hypothesis with multiple interrelating variables [40,41]. While there are many SEMs, here we used one of the more basic structures known as path analysis, which denotes a level of causality between a set of exogenous and endogenous variables [41,42]. Exogenous variables are those having no explanatory factors in the study but may have covariations. The endogenous variables are those attempting to be explained by the data and require a latent error variable. It must be noted that we will be discussing all associations in a matter of relationship dependency rather than causality due to the films’ complexity, and that the reasons for their success or failure cannot often be based on a singular aspect. While we know aspects of accuracy, anthropomorphism, environment, and self-agency can affect people’s perceptions of non-human primates [21], we set to explore which of these aspects had the strongest relationships to the media and their association with audience reception via gross box office profit and critical acclaim (Figure 1). Due to the restricted sample size, we used one indicator from each of the four constructs in the final models. The four constructs were: accuracy indicated by coloration, locomotion, range or natural vocalization; anthropomorphism indicated by the use of clothing, anthropomorphic vocalization, sign language, or unnatural locomotion; environment indicated by the primate’s status as wild, pet, captive or living in an anthro-like society; and agency indicated by whether the primate had agency or was used as a prop.

The general consensus on the use of SEMs states to maintain the stability of the parameter estimates in the model; a 10:1 ratio of ‘participants’ to parameters should be kept [42]. Therefore, we created multiple iterations of each model and presented the final models with the highest probability level shown by the chi-square test, the comparative fit index (CFI) closest to 1.000, the root-mean-square error of approximation (RMSEA) closest to 0.000, and the lowest Akaike information criterion (AIC) [40]. All the models used estimated means and intercepts to handle missing data with maximum likelihood for discrepancies. We allowed for non-positive definite sample covariance matrices but did not allow the attempted fit of unidentified models. The covariances supplied as the inputs were unbiased, but covariances were analyzed with maximum likelihood. We also used bias-corrected confidence intervals set to 0.95.

## 3. Results

### 3.1. Descriptive Statistics

The film list compilation resulted in 101 English-speaking films, which portrayed at least one primate, and debuted between 2000 and 2019. The films’ mean gross profit was 294,063,348.60 USD (SD = 33,130,056.49). The top three grossing films were *The Lion King* (2019), *Aladdin* (2019), and *Pirates of the Caribbean: Dead Man’s Chest* (2006), all grossing over 1 billion USD. The lowest grossing film was *Spymate* (2006), with a 45,007 USD gross profit. The average Rotten Tomatoes critics score was 55.91 out of 100 (SD = 2.61, range: 11–100). *Virunga* (2014) received the highest critic score while *Crocodile Dundee in Los Angeles* (2001) scored lowest, not including three films that did not have critic consensus scores. Live action was the most popular medium, followed by animation, CGI, and costumed primates. However, the use of live-action primates decreased over time as CGI use increased (Figure 2). Appearances in animation fluctuated by year, but on average stayed relatively steady over the 20-year period.

Of the films, 26 displayed multiple genera, resulting in a final sample size of 151 primates. We recorded 25 distinct genera within African apes, Asian apes, Central and South American monkeys, African and Asian monkeys as well as Prosimians accounting for 139 of the primates (Appendix A). The remaining primates consisted of one fossil primate along with generic apes and generic monkeys. Of the total 151 primates, 114 (75.50%) were shown engaging in some kind of human interaction. Of the primates observed, chimpanzees (*Pan troglodytes*) and capuchins (*Cebus* spp. And *Sapajus* spp.) featured most prominently, each appearing in 32 films; however, 85.50% of capuchins were depicted by live primates, whereas only 50.00% of chimpanzees were portrayed by live primates.

### 3.2. Path Analysis

From the exploratory models generated using the theoretical model (Figure 1), we produced six final models based on interactions among accuracy, anthropomorphism, environment, and agency indicators having the greatest relationship in predicting the medium’s use and its relationship to audience reception. We excluded the costumed medium from the final model analysis due to unfit models from low occurrences.

In the final models, the live-action primate medium was negatively related to both USD and the Rotten Tomatoes critic scores, indicating that films with live-action primates were more likely to profit less and be less acclaimed by critics. In the final models, predictors for the live-action medium and its relationship with audience reception were the same for gross profit and critic scores (Figure 3). The predictors of unnatural locomotion and agency held a negative significant relationship with the live-action medium, while the captive setting held a significant positive relationship to the medium (Table A2). The only significant relationship in the model held by primates depicted in their accurate range was the negative covariation with captive setting. The other three predictors held significant covariance with each other. Primates depicted engaging in unnatural locomotion were more likely to be depicted with some form of agency, and less likely to be depicted in a captive setting.

The animation medium had fit models, but there were no models in which animated primates had a significant relationship with worldwide gross profit or Rotten Tomatoes critic scores (Figure 3). The predicting factors for the highest probable model for the relationship between animated primates and gross profit worldwide were the same as those for CGI-depicted primates and gross profit worldwide. They also displayed the same pattern of significant association with the medium and covariates, but at slightly varied rates (Table A2). The predictive factors in the most probable model for animation in relation to critic scores were accurate coloration, unnatural locomotion, pet settings, and primates used as props (Model D). Accurate coloration had a significant negative predictive relationship with animation and unnatural locomotion had a positive predictive relationship with the medium. Unnatural locomotion held a significant positive covariation with primates in a pet setting, and a negative significant covariation with primates as props.

The models relating to primates portrayed by CGI indicated a significant positive relationship between the medium and both gross profit and critic reviews (Figure 3). However, unlike the models for the live-action medium, those of the highest probability for the CGI medium contained different predictors for the audience-reception types. The predictors for the most probable model representing the relationship between CGI and gross profit were natural-like vocals, clothing, set setting and agency (Model E). Of those, natural-like vocals and agency held a significant positive relationship, and primates in clothing held a significant negative relationship, with the medium. There was a significant positive covariance seen between clothing and agency, as well as clothing and primates in a pet setting. The most probable indicators for the relationship between CGI primates and Rotten Tomatoes critic scores were accurate locomotion, anthro-like vocals, pet setting and primates as props (Model F). Accurate locomotion was a significant positive predictor for the medium’s relationship with critic scores. Primates in pet settings and used as props held a negative predictive relationship with the CGI and critic scores relationship. Accurate locomotion held a positive significant covariation with primates in pet settings and a negative covariation with anthro-like vocals. Anthro-like vocals also held a negative covariation with primates in a pet setting.

## 4. Discussion

### 4.1. Accuracy

Since over half of the films included in this study were rated G or PG, one would hope that accuracy would be regularly employed for children, who often learn science concepts from entertainment [43]. However, accuracy indicators were only significant in four of the six predictive models, and the negative relationships with animation indicate that this format may experience some of the careless inaccuracies seen in children’s book illustrations [13]. While the lemurs in *Madagascar* (2005) have general appearances of different lemur species, inaccuracies are easy to acknowledge in other areas. For example, all lemur species appear together as one jungle society under the guidance of a male ring-tailed lemur (*Lemur catta*) and his advisor, an older aye-aye (*Daubentonia madagascarensis*). The most basic issues there are three-fold: lemurs live in a variety of habitats across Madagascar, ring-tailed lemurs live in female-dominant groups, and aye-aye are nocturnal and relatively solitary [44]. While the inaccuracies may be intentional for better storytelling and a result of anthropomorphism, it may cross the line of being enough anthropomorphism to generate concern and connection [11,45].

### 4.2. Anthropomorphism

Aspects of anthropomorphism were significant in five of six models. While anthropomorphism has shown to positively contribute to perceptions and conservation efforts [46], this relationship is fragile [11,47]. The two significant predictive indicators of primates performing unnatural locomotion and wearing clothing are particularly concerning. The indirect effect of unnatural locomotion for live-action primates regarding gross profit and critic scores signified that it is beneficial to show live primates moving in an unnatural manner. This creates possible welfare concerns. Even when AHA ‘No Animals Were Harmed’ guidelines are followed, these guidelines specify working conditions such as housing and hours preforming, but do not strictly regulate types of trained behaviors that may be stressful for a primate’s body and mind [27,28]. The other salient indicator of anthropomorphism was clothing. Our results showed that the lack of primates in clothing likely predicts higher gross profit for CGI, yet animation was positively linked to clothing. While primates in clothing is often discussed outside of research, Ross et al. [21] did not find the presence of chimpanzees in clothing to significantly impact perceptions of their threat status or pet suitability. However, their study specifically focused on live-action chimpanzee images. It is presumed that films using realistic CGI would not then add clothing as another aspect to generate in their images, as they are pursuing live-action realism; though, this would only be conjecture. Clothing on animated primates may be a welfare concern, but the use has not been seen to negatively affect conservation perceptions [21,23]. Further study is needed to determine if the suggestion by Root-Bernstein et al. [11] of having ‘just enough’ anthropomorphism supports positive perceptions and conservation behaviors.

### 4.3. Environmental

While clothing on primates may not be a substantial predictor of viewer perceptions, environmental context had significant effects on perceptions regarding the stability of wild populations [21,23]. Though the environmental context seemed to be a significant predictor in three of the six models, the relationship with a captive environment and live primates in film shows that they often appear in captive settings but receive poor reviews and less profit. Similarly, for CGI, primates in a pet or human companion environment had negative indirect effects on critic reviews. While the absence of human companions could be seen as positive for conservation, we suspect the relationship to be less of a reflection on how the primates are viewed, and possibly on how well the CGI technology is stitched into live-action scenes overall.

### 4.4. Agency

The delineation between the primate’s agency and use as a prop was the indicator that held a significant effect in five of the six models. However, agency and anthropomorphism are not synonymous despite the potential simplicity of ascribing agency to an animal that is highly anthropomorphized and questioning whether live-action animals are depicted as acting agents in their own lives. For example, in *Mary Poppins Returns* (2018) animated primates are in clothing and walking upright but merely set the scene of another adventure where Mary Poppins brings the children. Conversely, in *Pirates of the Caribbean: Dead Man’s Chest* (2006), Jack the monkey is a live-action capuchin whose only anthropomorphic characteristic is his pirate clothing. Jack’s agency stems from his routine-independent actions to change his situation, and at one point can even be seen leaving the human-centric action to find his previous, presumed dead, companion on his own accord. However, the relationship of agency and medium still leans toward live-action primates that are more likely used as props than acting agents in their narrative, whereas others in CGI and animation are more likely to have agency.

### 4.5. The Switch, Live Animals to CGI

Of course, accuracy, anthropomorphism, environment, and agency are viewed through the lens of the medium used. Though the degree of relation changes among the variables, live action was always significantly negatively related to the audience-reception indicators, CGI was always significantly positively related, and animation held no significant relationship. Regarding gross profit, the timeline could be a contributing factor. Over the 20-year timespan, the use of live-action primates gradually decreased, while CGI use increased. The gross profits of the films are not adjusted for the gradual inflation of the USD, which has grown by about 50.47% since 2000 [48]. Yet, well before 2000, films that were more technologically advanced still often gained more gross profit than others [15]. Time, however, does not affect the critics’ evaluation in a similar manner, and the medium’s relationship with critic reviews may be a little more surprising. It would appear that for CGI primates, crossing the uncanny valley threshold did not concern critic reviews. For live-action primates, it should be mentioned that the category included highly evaluated films such as *Virunga* (2014), *Jane* (2017), and *Project Nim* (2011), which earned Rotten Tomatoes critic consensus scores 100, 98 and 97, respectively, out of a possible 100. As these were all documentaries, they do not adhere to the same welfare concerns as films using live ‘actor’ primates. If these had been removed from the sample, though, then there may have been a stronger negative relationship between live-action primates and critic reviews.

The relationship of live-action primates with gross profit and critic reviews seemed to show that using primate ‘actors’ is not the most beneficial to film producers, especially if CGI is an alternative. Cost is often the main topic of discussion when debating using live animals or CGI technology, but it is not simple to calculate. Visual effects such as CGI are notoriously expensive, and animals are no exception, as seen when the *Game of Thrones* (2011-2019) creators cut the main character Jon Snow’s dire-wolf, choosing to use more of the special-effect budget on creating lifelike dragons [49]. However, the use of live animals on set can also be quite costly. Crystal the monkey, a capuchin, has appeared in numerous television and film programs, reportedly earning 180,000 USD in 2012, more than the average human actor [50]. However, this number does not exactly reflect what producers spend, considering even the most well-trained animal ‘actors’ can take a long time during film production, especially when they are not presenting the behaviors the director envisions [51]. Writing off using CGI animals in favor of live animals is not simply budgetary, especially when the results show films using CGI animals have higher gross profit. The *Aladdin* (2019) remake, for example, featured a fully CGI capuchin as Abu in a similar role where the animal ‘actor’ Crystal the monkey would be cast. While production reportedly cost 183 million USD, it grossed over 1 billion USD [52]. The perception of these animals is still unclear, though, despite CGI seeming to be a promising alternative to live exotic animals while also sparing individuals’ welfare concerns regarding animal use on a set. Though Abu is portrayed entirely using CGI stitched into live-action scenes, his appearance as a pet monkey may retain the perception implications seen by Ross et al. [21] with live chimpanzees and their compatibility as pets.

### 4.6. Consistency of Animation

Unlike live-action and CGI depictions, animation had no significant effects on gross profit or critic reviews. However, this does not exclude it from holding an important role in how the public constructs their primate views. The relatively steady rate at which animated movies with primates were released over the 20 years suggests their depiction in the medium is relatively constant. Even considering that the animated films sampled were not overwhelming blockbusters or critic darlings, animations often have a wide audience reach and a memorable impact on viewers [10,53]. The animated depictions of primates had strong relationships with accuracy and anthropomorphism indicators, but were more likely to be anthropomorphized and less likely to depict accurate coloration or locomotion. However, children as young as five years are able to distinguish between fantasy and reality in films [54], as well as to distinguish animations as not ‘real’ [55]. This may help explain why major animated films have been shown to generate interest in different species [14], but do not significantly influence the pet trade [24].

### 4.7. A Changing Industry and Changing Research

It must be noted that many of these films originate and are produced by some of the same production companies regardless of medium or different portrayal predictors per medium, and each relates to audience reception differently. Walt Disney Pictures, for example, produced over 20% of the films observed, including the top three grossing films observed, and portrayed primates in each medium. Disney is also one of the most scrutinized production companies regarding the use and portrayal of animals [8,53,56,57]. Disney’s influence in film and entertainment is not hard to feel, having a worth of approximately 238 billion USD and becoming the world’s largest media conglomerate in 2019 [58]. Disney has become the film industry’s posterchild of the highly curated images and understands how important image is to perception and behavior. Their animation could often be described as biophilic, but unrealistic and over-fantasized [53,56]. In 2012, Disney announced a policy in which they stated the company’s general practice would be to no longer use live exotic animal ‘actors’, with occasional exceptions. One of those exceptions is represented by the capuchin in the last instalment of their Pirates of the Caribbean franchise, *Dead Men Tell No Tales* (2017). However, primates are given special attention in this policy, as it states that there will be no exceptions for apes, or ‘large primates’ such as baboons and macaques. While this policy may merely be a reflection of the film industry recognizing the well-studied repercussions of using live primates in entertainment [21,22,23], there is less understanding of the effects of the growing popularity of CGI and the steady use of animation. If Disney is an example of how the industry responds and changes to research and public knowledge, then it is imperative that research continue in these areas to keep up with the ever-growing industry as we continue to see the rise in CGI and the steadfast use of animation.

## 5. Conclusions

### 5.1. Animation

Overall, the relationship between gross profit and critic scores per medium indicates an important point. While animation did not have significant effects on gross profit or critic reviews, it has remained constant in its use to represent primates in film. Thus, animators’ use of accuracy and anthropomorphism in their portrayal of primates should be studied further for their effects on public perceptions.

### 5.2. Live Action v. CGI

The use of live-action primates has not especially benefitted film producers, and with concerns of animal welfare, trade, and public perceptions, the decision for film producers not to use live-primates in their films should be an easy one. CGI primates, alternatively, resulted in positive audience reception via gross profit and critic scores. Although CGI does not hold the same welfare concerns, due to the increasing use of this medium and its use to simulate live action, further quantitative and qualitative study should be performed to determine if these portrayals could affect public perception and pet trade. Since it has been shown that different images of live-action primates influence perceptions of conservation status [21,23], and since CGI often attempts to mimic reality, the CGI portrayals of primates should be examined to understand if people interoperate CGI portrayals in similar ways to live action.

### 5.3. Moving Forward

Here we showed that the use of primates in animation has remained steady, and those in live-action films have decreased throughout the past two decades and occur in less popular films, especially compared to CGI portrayals. However, further investigation into how people interoperate the difference in medium portrayals is imperative to continue to understand how this multibillion-dollar industry impacts conservation efforts.

## Figures and Tables

**Figure 1 animals-12-01576-f001:**
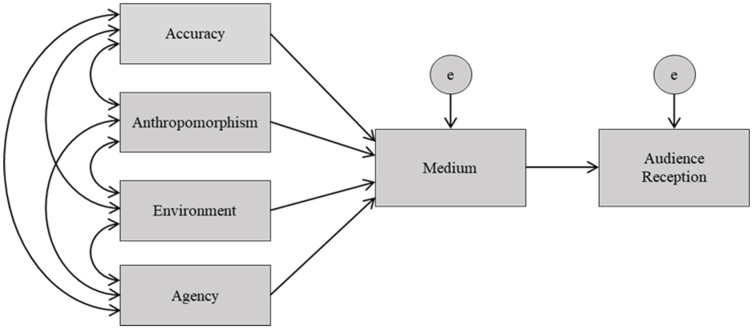
SEM Theoretical Model. Each line indicates a parameter set on the model. Double-headed arrows represent a covariance. Single-headed arrows represent direct effects. Rectangles are the observed variables. Circles are unobserved, latent variables where ‘e’ is the residual error.

**Figure 2 animals-12-01576-f002:**
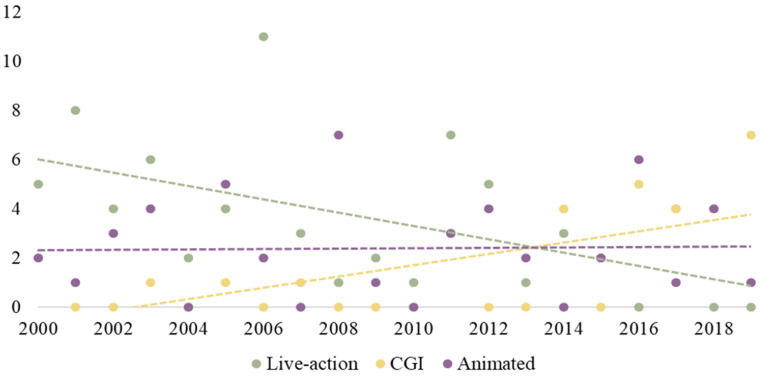
Annual frequency of medium use. Each point represents the number of recorded primate occurrences in each medium, per year. Dashed lines represent the average trends overtime for medium depiction.

**Figure 3 animals-12-01576-f003:**
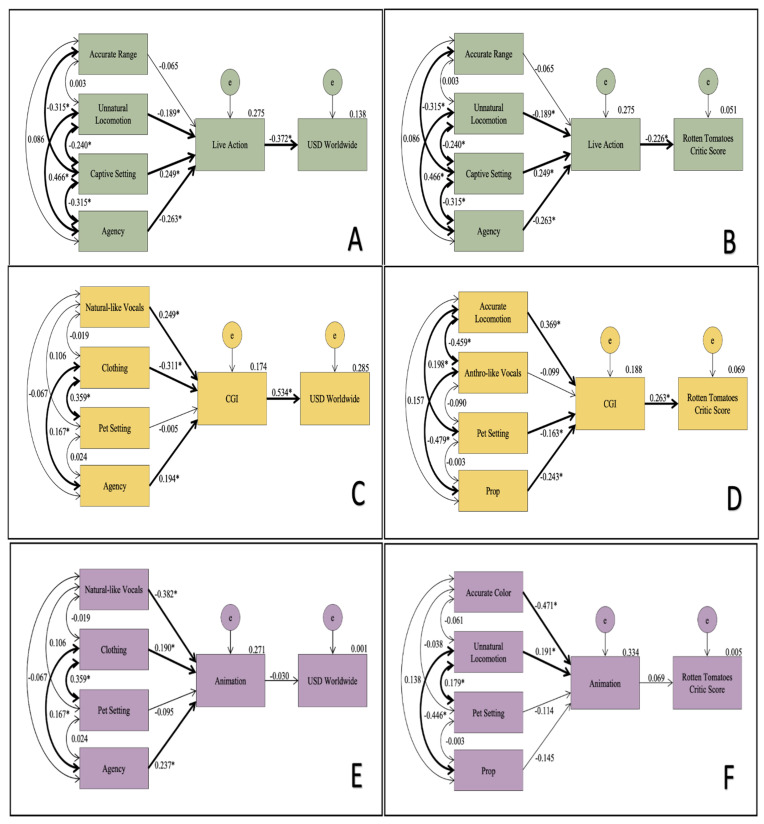
Path-analysis models for live-action, CGI and animation medium. Model **A** (χ^2^(4) = 1.693; *p* = 0.792; CFI = 1.000; RMSEA = 0.000) represents the most probable factors to predict the use of live-action primates based on the relationship with the film’s gross profit worldwide in USD. Model **B** (χ^2^(4) = 2.067; *p* = 0.723; CFI = 1.000; RMSEA = 0.000) represents the most probable factors to predict the use of live primates based on the relationship with the film’s critic consensus scores from Rotten Tomatoes. Model **C** (χ^2^(4) = 3.387, *p* = 0.495; CFI = 1.000; RMSEA = 0.000) represents the most probable factors to predict the use of animation based on its relationship with the film’s gross profit worldwide in USD. Model **D** (χ^2^(4) = 3.608, *p* = 0.462; CFI = 1.000; RMSEA = 0.000) represents the most probable factors to predict the use of animation based on its relationship with the film’s critic consensus scores from Rotten Tomatoes. Model **E** (χ^2^(4) = 1.823; *p* = 0.768; CFI = 1.000; RMSEA = 0.000) represents the most probable factors to predict the use of CGI based on its relationship with the film’s gross profit worldwide in USD. Model **F** (χ^2^(4) = 2.026; *p* = 0.731; CFI = 1.000; RMSEA = 0.000) represents the most probable factors to predict the use of CGI based on its relationship with the film’s critic consensus scores from Rotten Tomatoes. Bolded arrows, paired with starred standardized regression weight or covariance, indicate a statistically significant relationship.

## Data Availability

The data presented in this study are available in Appendix A.

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
