# Peer review of "The Use of Live Action, Animation, and Computer-Generated Imagery in the Depiction of Non-Human Primates in Film"

_animals, 2022, doi:10.3390/ani12121576_

Round 1

Reviewer 1 Report

Review of Animals

Alexandra Martinez et al.

"The use of live-action, animation and computer-generated imagery in the depiction of non-human primates in film"

This was an interesting article on the use of primates in film, with implications for conservation.  I think it will be of general interest to a large readership.  The paper was well written, but I did find a number of places where words were missing or there were other grammatical issues.  I've listed those below, but it could use a careful proofread.  

The only significant issue that I think needs addressing are Figures 3 and 4.  They aren't very useful.  It would be much more impactful to show your data in a more typical format - bar charts for example.  

Line 17:  "Critical consensus" is a bit vague

Line 32:  Should be animations 'have' increased not has

Line 55:  Clarify that this was a real horse that went over the cliff

Line 104:  "Not only is this concern for the individual psychological welfare" needs to be reworded.  Please change 'this' to something more descriptive.  I think what you're saying is the barred teeth display may indicate that the animal is in distress. Should be 'of' concern.

Line 105:  Should be "the" likelihood of trade

Line 126: Should be 'are given' not 'are to given'

Line 130: "the other conservation concerns" is too vague.

Line 137: 'show the have effect' need to be reworded to 'shown to have an effect'

Line 143:  'and characterize' doesn't match the part of speech with rest of sentence.

Line 149:  'online streaming exclusive films' - unclear what you mean?  I guess you to add 'of' exclusive films?

Line 132:  Should be 'had' not 'held'

Line 339:  'nationwide' should be 'across Madagascar'

Line 340:  I think you can also say that lemurs don’t live in mixed-species societies.

Line 347:  Should be 'are' particularly concerning

Line 381: "Mary Poppins brings the children" needs to be reworded.  I guess you could say "where" Mary Poppins... Or restructure the sentence.

Line 409:  Should be "Cost if often the main topic" not 'in the main topic'

Line 416:  Should be 'what producers spend'

Line 440:  Should be 'have "been" shown' and 'but do not significantly influence the pet trade'

Line 448: delete 'nearly'

Line 449: specify 'the world's largest media conglomerate'

Appendix B can go in supplementary materials

Author Response

Alexandra Martinez et al.

"The use of live-action, animation and computer-generated imagery in the depiction of non-human primates in film"

This was an interesting article on the use of primates in film, with implications for conservation.  I think it will be of general interest to a large readership.  The paper was well written, but I did find a number of places where words were missing or there were other grammatical issues.  I've listed those below, but it could use a careful proofread.  

We thank the reviewer for the very kind words, and we appreciated the suggestions provided to ameliorate this piece of work. We have considered all of them.

The only significant issue that I think needs addressing are Figures 3 and 4.  They aren't very useful.  It would be much more impactful to show your data in a more typical format - bar charts for example.  

We would like to point out that Fig. 3 and 4 are the representation of SEMs and they should be presented. They cannot be replaced by bar charts as they do not have the same meaning.

Line 17:  "Critical consensus" is a bit vague
Changed to critic score

Line 32:  Should be animations 'have' increased not has

changed

Line 55:  Clarify that this was a real horse that went over the cliff

added

Line 104:  "Not only is this concern for the individual psychological welfare" needs to be reworded.  Please change 'this' to something more descriptive.  I think what you're saying is the barred teeth display may indicate that the animal is in distress. Should be 'of' concern.

Edited

Line 105:  Should be "the" likelihood of trade

added

Line 126: Should be 'are given' not 'are to given'

changed

Line 130: "the other conservation concerns" is too vague.

changed

Line 137: 'show the have effect' need to be reworded to 'shown to have an effect'

changed

Line 143:  'and characterize' doesn't match the part of speech with rest of sentence.

changed

Line 149:  'online streaming exclusive films' - unclear what you mean?  I guess you to add 'of' exclusive films?

That means that some films are exclusive of online streaming platforms 

Line 132:  Should be 'had' not 'held'

The line number does not match

Line 339:  'nationwide' should be 'across Madagascar'

changed

Line 340:  I think you can also say that lemurs don’t live in mixed-species societies.

That is generally true but there are some specific cases when individuals of one species live with a group of another species even in the wild, so we prefer to avoid to add this statement. See for example: Eppley, T. M., Hall, K., Donati, G., & Ganzhorn, J. (2015). An unusual case of affiliative association of a female Lemur catta in a Hapalemur meridionalis social group, Behaviour152(7-8), 1041-1061. 

Line 347:  Should be 'are' particularly concerning

Changed

Line 381: "Mary Poppins brings the children" needs to be reworded.  I guess you could say "where" Mary Poppins... Or restructure the sentence.

Changed

Line 409:  Should be "Cost if often the main topic" not 'in the main topic'

Changed

Line 416:  Should be 'what producers spend'

Changed

Line 440:  Should be 'have "been" shown' and 'but do not significantly influence the pet trade'

Changed

Line 448: delete 'nearly'

Deleted

Line 449: specify 'the world's largest media conglomerate'

Added

Appendix B can go in supplementary materials

We think Appendix B is important as it shows the whole model outputs. Since MDPI allows the attachment of appendices in the main text, we would like to keep it in the main text but we can move it if the editor prefers.

Reviewer 2 Report

Please see uploaded doc

Author Response

General remarks:
This is a very interesting article with an important analytical message for the film industry. The first impression one gets from the article is that it is investigating public impressions and perceptions of using primates in films, however, throughout the reading it was not exactly clear to me if this was investigated or if the perception data is solely based upon film critics. My advice would be to clarify and elaborate on this matter already in the abstract and introduction to explain what the goal was – whose perception of primates in movies – is the target of this study.

We thank the reviewer for the very kind words, and we appreciated the suggestions provided to ameliorate this piece of work. We have considered all of them.

In the conclusion it is becoming clearer through the following statement:
“Thus, animators’ use of accuracy and anthropomorphism in their portrayal of primates should be studied further for their effects on public perceptions.”
This is an important remark and a very interesting and crucial topic for future assessments, but it would be nice if the authors specify the perception audience for this study at an early stage.

We have now added this information in the introduction

Although, it is not the focus of the article. One cannot help to become curious about what are the applied outcomes from these results. Do the authors see potential regarding how this generated knowledge can be used to influence and advice film industry and lawmakers? If appropriate within the scope of the journal, maybe briefly elaborate on the implications in the discussion section.

We agree that this is important, and we would like to refer to the section A Changing Industry and Changing Research. 

Detailed comments:
L60: The Sentence sounds a bit awkward starting with Wells reasons, maybe clarify to something like The author Wells reasons…

Changed

L137: Typo error: “all have shown the have effect on how people understand the primates” – Please correct to “all have shown to influence …”

Changed

L147: Please clarify what type of films? Only blockbusters or also short films? Commercials?

Added

L194-L195: Please somewhere specify who did the ratings and how they were scored in terms of subjectivity.

Added

L330: Please remind the reader again, what these shortenings stand for.

Added

Reviewer 3 Report

This is an interesting and important paper (and one of the most fun to read in a long time!). The authors make an excellence job trading in between two distinct disciplines. They make an excellent job connecting and explaining the film industry and primate conservation efforts. Their analyses are thorough, well implemented and, critically, revealing of how choices in the entertainment industry can impact primates, either directly (welfare) or indirectly (audience's perceptions). Results have the potential hold implications on filming decision in the future, both for producers' own benefit and primate conservation and welfare.

line 59-63: Please rewrite/rephrase/clarify/elaborate for clarity (e.g., what does "draw the abundance of frames needed to produce a film" mean?).

77: Please spell out the CGI acronym on it's 1st appearance in text (excl. Abstract)

86-87: Please rephrase for clarity ("Yet,...")

93: "Than others" what? What does "others" stand for? Animals?

98: What does "whom" refer to? Human handles/traders/trainers?

239: Close brackets after 2006

312: Use past tense

336-343: Spot on!

348-353: Spot on!

Discussion: Please divide the discussion into sub-headings, too much continuous text currently.

Conclusion: Please divide into sub-headings per medium for ease of readership, specially for lay and/or entertainment industry readers.

Author Response

This is an interesting and important paper (and one of the most fun to read in a long time!). The authors make an excellence job trading in between two distinct disciplines. They make an excellent job connecting and explaining the film industry and primate conservation efforts. Their analyses are thorough, well implemented and, critically, revealing of how choices in the entertainment industry can impact primates, either directly (welfare) or indirectly (audience's perceptions). Results have the potential hold implications on filming decision in the future, both for producers' own benefit and primate conservation and welfare.

We thank the reviewer for the very kind words, and we appreciated the suggestions provided to ameliorate this piece of work. We have considered all of them.

line 59-63: Please rewrite/rephrase/clarify/elaborate for clarity (e.g., what does "draw the abundance of frames needed to produce a film" mean?).

Rephrased

77: Please spell out the CGI acronym on it's 1st appearance in text (excl. Abstract)

Added

86-87: Please rephrase for clarity ("Yet,...")

Rephrased

93: "Than others" what? What does "others" stand for? Animals?

Added

98: What does "whom" refer to? Human handles/traders/trainers?

Changed

239: Close brackets after 2006

Added

312: Use past tense

Changed

336-343: Spot on!

348-353: Spot on!

Thanks!

Discussion: Please divide the discussion into sub-headings, too much continuous text currently.

Thanks for the suggestion

Conclusion: Please divide into sub-headings per medium for ease of readership, specially for lay and/or entertainment industry readers.

Thanks for the suggestion